# Diabetic Complications and Oxidative Stress: A 20-Year Voyage Back in Time and Back to the Future

**DOI:** 10.3390/antiox10050727

**Published:** 2021-05-05

**Authors:** Carla Iacobini, Martina Vitale, Carlo Pesce, Giuseppe Pugliese, Stefano Menini

**Affiliations:** 1Department of Clinical and Molecular Medicine, “La Sapienza” University, 00189 Rome, Italy; carla.iacobini@uniroma1.it (C.I.); martina.vitale@uniroma1.it (M.V.); stefano.menini@uniroma1.it (S.M.); 2Department of Neurosciences, Rehabilitation, Ophthalmology, Genetic and Maternal Infantile Sciences (DINOGMI), Department of Excellence of MIUR, University of Genoa Medical School, 16132 Genoa, Italy; pesce@unige.it

**Keywords:** advanced glycation end-products, antioxidants, diabetes, hyperglycemia, methylglyoxal, mitochondrial dysfunction, nuclear factor erythroid 2-related factor 2, reactive carbonyl species, reactive oxygen species, Warburg effect

## Abstract

Twenty years have passed since Brownlee and colleagues proposed a single unifying mechanism for diabetic complications, introducing a turning point in this field of research. For the first time, reactive oxygen species (ROS) were identified as the causal link between hyperglycemia and four seemingly independent pathways that are involved in the pathogenesis of diabetes-associated vascular disease. Before and after this milestone in diabetes research, hundreds of articles describe a role for ROS, but the failure of clinical trials to demonstrate antioxidant benefits and some recent experimental studies showing that ROS are dispensable for the pathogenesis of diabetic complications call for time to reflect. This twenty-year journey focuses on the most relevant literature regarding the main sources of ROS generation in diabetes and their role in the pathogenesis of cell dysfunction and diabetic complications. To identify future research directions, this review discusses the evidence in favor and against oxidative stress as an initial event in the cellular biochemical abnormalities induced by hyperglycemia. It also explores possible alternative mechanisms, including carbonyl stress and the Warburg effect, linking glucose and lipid excess, mitochondrial dysfunction, and the activation of alternative pathways of glucose metabolism leading to vascular cell injury and inflammation.

## 1. Introduction

Hyperglycemia is the primary etiologic factor of vascular disease associated with diabetes mellitus (DM) and accounts for most of the morbidity and mortality associated with both type 1 DM (T1DM) and type 2 DM (T2DM) [1]. Diabetic retinopathy and nephropathy are the leading contributors to adult blindness and renal failure, and, in the diabetic population, accelerated atherosclerosis increases the risk of myocardial infarction, stroke, and lower limb amputation [1,2]. Unfortunately, even strict glycemic control-regimen does not completely prevent the development and progression of long-term complications. While early intensive glycemic control was shown to provide protection from diabetic microvascular disease [3], the effect of tight glycemic control was rather disappointing in terms of prevention of macrovascular complications if not established early in the course of the disease [4]. This evidence supports the view that the pathogenesis of the chronic complications of DM is complex and multifactorial in nature. Accordingly, in addition to hyperglycemia, comprehensive risk factor modification including obesity, dyslipidemia, hypertension, and platelet activation is mandatory in the medical management of patients with DM [1]. Moreover, for educational, motivational, and economic reasons, optimal metabolic control of DM is challenging to achieve in a real-world setting. Accordingly, the rate of diabetic complications still represents a major concern for diabetic patients and global healthcare [5]. Because of this, novel disease modifying therapies targeting the impact of glucose and lipid toxicity on blood vessels are desperately needed.

During the last century, many hypotheses have been proposed to explain the origin of diabetic complications. These include the advanced glycation end-products (AGEs) (also known as Maillard hypothesis) [6,7], oxidative stress [8,9,10,11], pseudohypoxia [11,12], true hypoxia [13], carbonyl stress [14], polyol pathway activation [15], increased protein kinase C activity [16], disturbances in lipoprotein metabolism [14,17,18], and altered cytokine [19] or growth factor [20] expression. In part, the number and variety of the hypotheses put in place indicated uncertainties in the understanding of the pathogenesis of diabetic complications: they could reflect either the activation of tissue-specific stress response pathways, with different pathogenic mechanisms acting in different tissues, or a common underlying pathological mechanism. Many agreed that these hypotheses are mechanistically intertwined and overlap with each other. In particular, the idea that oxidative stress could act as a common thread between the various hypotheses was widely shared. In fact, oxidative stress may promote AGE formation, and, vice versa, AGEs may induce inflammatory cytokine and growth factor expression through activation of redox signaling pathways; increased polyol pathway activity may lead to oxidative stress; hypoxia induces glycolysis and perpetuates inflammation through radical oxygen species (ROS) formation; carbonyl stress is both a consequence and a trigger of oxidative stress; and so on. Accordingly, at the beginning of the new century, a unifying hypothesis on the pathobiology of diabetic complications was formulated, with ROS proposed as the initial instigators of most of the major pathways previously shown to be involved in the onset and progression of diabetic complications [21]. The unifying hypothesis has quickly become a cornerstone of diabetes research, capable of drawing the attention of the vast majority of investigators in the field of diabetic complication to date. Unfortunately, the large number of promising results from preclinical animal models have failed to translate from bench to bedside, as several antioxidants tested in human trials have yielded disappointing results showing, at best, a mild protective effect against diabetic complications. What is more, in recent years, experimental evidence has emerged suggesting that diabetic complications can arise in the absence of increased ROS production.

Following a brief recapitulation of the biological background underlying the single unifying hypothesis proposed by Brownlee and colleagues [21], this review summarizes the most relevant literature of the last twenty years on the main sources and mechanisms of ROS generation in DM and their role in the pathogenesis of diabetic complications. Then, the evidence in favor and against a primary role for oxidative stress and, in particular, the state of the art regarding the issue of whether or not ROS overproduction represents the initial trigger of the biochemical abnormalities induced by hyperglycemia is discussed. Finally, possible alternative mechanisms linking glucose and lipid excess with mitochondrial dysfunction and the activation of pathogenic pathways leading to vascular cell injury and end-organ damage are presented.

## 2. Mitochondrial ROS as the Underlying Common Pathogenic Mechanism of Glucotoxicity: A Single Unifying Mechanism for Diabetic Complications

Historically, the observation that not all cells and tissues are prone to hyperglycemia-induced damage and chronic complications brought the attention to differences about how different cells handle excess extracellular glucose. In kidney cells such as podocytes, mesangial cells, and tubular cells, as well as endothelial cells and cells of the immune system. Moreover, in cells of the neuron system, such as neurons and glia, glucose enters via facilitated diffusion through the insulin-independent glucose transporter 1 [22,23]. Therefore, in these cells, hyperglycemia can lead to intracellular excess glucose, forcing the cell to metabolize it and dispose of metabolic intermediates, which results in unfavorable biochemical consequences. This explains why the kidney, the cardiovascular system, the eye, and the nervous system are typical target organs of DM.

At the end of the last century, the bulk of publications about the mechanisms underlying DM-induced vascular damage focused on four major pathways branching from glycolysis: increased flux of glucose and its downstream metabolites through the (1) sorbitol/polyol and (2) hexosamine pathways; (3) augmented intracellular formation of AGEs and expression of the receptor for AGE (RAGE); and (4) activation of protein kinase C (PKC) isoforms [24,25]. Interventions aimed at blocking only one of these pathways, i.e., with specific inhibitors of aldose reductase activity, hexosamine pathway flux, AGE formation and RAGE ligand binding, and PKC activation, were shown to ameliorate various hyperglycemia-induced abnormalities in preclinical models [6,24,26,27,28,29,30] but yielded inconclusive results in clinical studies and humanized mouse models [31,32]. Together with the observation that the activation of all the four pathogenic pathways was rapidly reversed when euglycemia is restored, this led to the reasoning that these processes are interconnected or might have a common underlying causal mechanism. In 2000, Brownlee and colleagues proposed for the first time a single unifying mechanism for diabetic complications, demonstrating that the four pathways mentioned above stem from a single process induced by hyperglycemia: mitochondrial overproduction of ROS [21].

Normally, for the great majority of molecular oxygen (O_2_) processed in the mitochondria, the conversion to water involves the four-electron reduction of O_2_ by hydrogen. This is a finely coordinated process, in which the electrons from the electron donors NADH and FADH2 are transferred through Complexes I–IV of the electron chain and protons are extruded outwards into the intermembrane space, generating a proton gradient (Figure 1).

Flowing down their gradient and back into the matrix, protons drive the synthesis of ATP by passing through the adenosine triphosphate (ATP) synthase (Complex V). However, 1–4% of O_2_ conversion physiologically leads to formation of incomplete reduced molecular O_2_, generating highly reactive and short-lived intermediary products, called ROS [33,34,35]. The primary ROS produced in mitochondria are superoxide anion, which derives by one-electron reduction of O_2_, and hydrogen peroxide (H_2_O_2_), which represents the product of superoxide dismutation catalyzed by superoxide dismutase (SOD) enzymes [36]. Generation of mitochondrial ROS can rise in response to a reduction of O_2_ supply, an increase in substrate supply, or both [33,35]. In hyperglycemia, the enhanced glucose flux though glycolysis results in more glucose-derived pyruvate being oxidized in the mitochondrial tricarboxylic acid (TCA) cycle, increasing the flux of electron donors into the electron transport chain and the voltage gradient across the inner mitochondrial membrane. When a critical point of the voltage gradient is reached, the electron transfer inside Complex III is blocked [37], causing the electrons to back up to coenzyme Q, which transfers the electrons one at a time to O_2_, thereby generating superoxide [38].

Regarding the mechanism linking superoxide overproduction to the activation of the four pathways mentioned above, it was proposed that hyperglycemia-induced ROS generated in mitochondria activates the nuclear DNA-repair enzyme poly(ADP-ribose) polymerase (PARP) by inducing DNA strand breaks. Once activated, PARP modifies the key glycolytic enzyme glyceraldehyde 3-phosphate dehydrogenase (GAPDH) with polymers of ADP-ribose, causing a reduced activity of this enzyme [39] (Figure 2).

As a result, the levels of the glycolytic intermediates that are upstream of GAPDH increase. These include the triose phosphate glyceraldehyde 3-phosphate (GA3P), which is involved in the generation of methylglyoxal (MGO) and MGO-derived AGEs, and the production of diacylglycerol (DAG), a physiologic activator of PKC. Further upstream, the glycolytic metabolite fructose 6-phosphate and glucose itself also level up, resulting in an increased activity of the hexosamine and polyol pathways, respectively [39]. The finding that inhibiting GAPDH activity by antisense DNA resulted in the activation of each of the major pathways induced by hyperglycemia [39] provided further evidence for a critical role of GAPDH inhibition in hyperglycemia-induced cellular damage.

From these results, it appeared that mitochondrial superoxide production is required for the initiation of hyperglycemia-induced oxidative stress and damage. However, before and after Brownlee’s findings, growing evidence has accumulated that several other nonmitochondrial ROS production pathways may be activated by hyperglycemia and glucose metabolites. These include redox reactions between reducing sugars and protein amino groups, NADPH oxidases (NOXs), uncoupled endothelial nitric oxide synthase (NOS), etc. Moreover, at least 10 other ROS-producing sites have been identified in mammalian mitochondria in addition to that identified by Brownlee and coworkers [40]. Whether these additional sources of ROS are directly activated by hyperglycemia or merely reflect and amplify the original damaging effect of hyperglycemia is still matter of debate.

## 3. Non-Mitochondrial ROS Sources and their Role in Diabetic Complications

Irrespective of the debate on whether mitochondrial ROS overproduction is the initial instigator or an adjunct player in the pathogenic process, oxidative stress has been suggested to play a causal role in DM-related end-organ complications [40,41,42], including diabetic nephropathy (DN), retinopathy (DR), heart attack, and stroke, which affect 27.8%, 18.9%, 9.9%, and 6.6% of the diabetic population, respectively [43]. Collectively, these diabetic sequelae, including DN [44], represent major causes of morbidity and mortality in DM patients and are only partly prevented by glucose-lowering agents.

As mentioned above, besides hyperglycemia, several other stressors including dyslipidemia, AGEs, inflammation, and upregulation of the renin-angiotensin system can contribute to the harmful production of ROS [45]. Moreover, in addition to mitochondria, other cellular ROS generating pathways, including cytosolic NOXs [25], xanthine oxidase (XO) [46], uncoupled endothelial NOS [47], myeloperoxidase (MPO) [48], cycloxygenase-2 [49], and endoplasmic reticulum stress [50] (Figure 3) also contribute to the pathogenesis of diabetic complications, including DN [51], neuropathy [52], and atherosclerosis [48].

Importantly, the sources of ROS and their mechanisms of action vary with respect to different cellular sites and tissue types, sometimes even qualitatively [45]. In addition, each type of ROS (i.e., superoxide, H_2_O_2_, and hydroxyl radical) and oxidant species (i.e., nitric oxide, peroxynitrite, and HOCl) may play specific pathological roles depending on the organ affected. All these specificities regarding the sources and types of ROS and other oxidant species need to be carefully dissected and considered in the perspective of drug discovery [46]. Accordingly, clinical approaches targeting specific sources and types of cytotoxic oxidants in DM are ongoing (see below) or foreseeable in the near future.

Along with mitochondrial generation of superoxide, NOX enzymes appear to be the most important sources of ROS in DM [42,53,54,55] and have been implicated in the pathogenesis of diabetic micro- and macrovascular disease [25]. NOXs are tissues-specific complexes that produce ROS as second messenger. Unlike other enzymatic sources which have other primary functions and ROS represent a by-product of their normal activity, NOXs are the only enzyme family known to generate ROS as its sole function [56]. Seven members of the NOXs family, i.e., five NOX members (NOX1–5) and two dual oxidases formerly known as thyroid oxidases, have been identified so far. In humans, all seven oxidases are expressed, and each NOX isoform has a unique tissue expression pattern and specific regulatory mechanisms [56,57]. Of all isoforms, the most relevant ones in the pathophysiology of vascular complications are NOX1 and NOX5, which generate superoxide, and NOX4 (formerly termed Renox), which has a propensity for predominant H_2_O_2_ production. In particular, ROS derived from these NOX isoforms are thought to participate in the pathogenesis of atherosclerosis by favoring hypertension, inflammation, and a prothrombotic state, either directly or by reducing the availability of the vasoprotective molecule nitric oxide [41,58]. However, the specific role of NOXs in diabetic complications appears more complex than simply increasing ROS production. In macrovascular disease, one of the most prominent hypotheses to explain the increased cardiovascular risk conferred by DM is that hyperglycemia accelerates the atherogenic process by increasing ROS production. Consistently, NOX1-derived superoxide was shown to accelerate diabetic vasculopathy in preclinical models [41] and NOX5-derived superoxide has been associated with unstable cardiovascular disease in humans [59]. Conversely and surprisingly, this does not account for NOX4-derived H_2_O_2_, which was found to exert antiatherosclerotic effects by actively regulating smooth muscle cell responses in diabetic mice [60]. Differently from vascular wall, both NOX4-derived H_2_O_2_ and NOX5-derived superoxide seem to favor DN by triggering renal and glomerular hypertrophy, kidney fibrosis, and albuminuria [61,62]. In particular, H_2_O_2_ generated by NOX4, seems to increase directly the expression of profibrotic markers and vascular endothelial growth factor (VEGF) [61] and to induce glomerular hyperfiltration possibly by mediating endothelial NOS uncoupling and interfering with the nitric oxide/NOS axis in glomerular cells [63]. NOX4 was also involved in DM-associated inflammation, as its deletion reduced renal macrophage infiltration and downregulated monocyte chemoattractant protein-1 (MCP-1) and nuclear factor-κB (NF-кB) signaling, two important mediators of inflammation [61]. In addition, ROS derived from NOX4 and NOX1 have been involved in DR. H_2_O_2_ produced by NOX4 was demonstrated to induce blood–retina barrier breakdown by increasing VEGF expression [64], whereas superoxide originated from NOX1 has been involved in retinal cell damage and death [65]. In chronic diabetic neuropathy, ROS overproduction resulting from the crosstalk between liver X receptor and NOX4 was recently identified as a promising therapeutic approach to preserve Schwann cell integrity in diabetic mice [66]. In addition to genetic targeting, pharmacologic blockade of NOX4 and NOX1 showed promising results in preventing renal disease and atherosclerosis in preclinical models [41,61], but failed to reduce albuminuria in a short-term Phase II trial in patients with T2DM (ClinicalTrial.gov registration No. NCT02010242). The facts that none of the currently available NOX inhibitors is isoform specific [57] and that the different isoforms may have qualitatively opposing effects [46] could possibly explain this poor outcome. However, another trial is currently investigating the anti-albuminuric effect of a NOXs inhibitor in patients with T1DM and DN (ClinicalTrial.gov registration No. U1111-1187-2609).

The heme protein myeloperoxidase (MPO) is a major constituent of neutrophils with potent microbicidal and detoxifying activities [67] and is also expressed in monocytes, playing an important role in phagocytosis [68]. MPO shows peroxidase and haloperoxidase activities and converts NOX2-derived H_2_O_2_ in higher reactive species such as hypochlorous acid and peroxinitrite in the presence of chloride anion and nitrite, respectively, thus representing an additional source of peroxinitrite besides the canonical interaction between nitric oxide and superoxide (Figure 3) [69]. Some evidence exists that MPO levels are associated with the development of T2DM and its complications, including atherosclerosis [49] and DN [70]. Consistently, MPO activity and phagocytic cell-derived products of the oxidative burst, such as H_2_O_2_ and chloride, have been shown to trigger the formation of the reactive carbonyl species (RCS) acrolein, 2-hydroxypropanal, and glycoaldehyde [70]. Together with RCS derived from oxidative and non-oxidative metabolism of glucose and lipids [71], these RCS contribute to tissue accumulation of their final protein adducts, collectively known as advanced glycation end-products (AGEs). In particular, the α-hydroxyaldehyde formed from L-serine by neutrophilic MPO activity is able to form cross-links and generate Nε-(carboxymethyl)lysine (CML), a major AGE found in vivo that mainly derives from glyco- and lipoxidation reactions [72,73]. Therefore, ROS and other MPO-derived reactive compounds may promote the development and progression of diabetic complications by contributing to the accumulation of AGEs [74].

Xanthine oxidase (XO) is an oxidoreductase enzyme that generates ROS such as superoxide and H_2_O_2_ during oxidation of hypoxanthine to xanthine and further conversion of xanthine to uric acid [56]. Similar to MPO, elevated XO activity is associated with the development of T2DM and correlates with end-organ damage in experimental models and in humans [52,53]. Therefore, selective XO inhibitors such as allopurinol and febuxostat, which are used to lower uric acid level and reduce the risk of a gout flare in gout patients, may represent new drug candidates for the treatment of DM-related organ injury because of their ROS-lowering effects. Accordingly, allopurinol has been tested in a clinical trial for the treatment of DN with a focus on preventing glomerular filtration rate loss and coronary artery disease in T1DM (ClinicalTrial.gov registration No. NCT02017171); however, results have been disappointing [75].

Awaiting the results of ongoing clinical trials on new drug candidates targeting specific sources and types of ROS, it should be taken the opportunity to identify and fill possible knowledge gaps and misconceptions about the role of ROS in diabetic complications. Considering that each of the relevant therapeutic targets discussed in this section probably contributes to form a ROS-signaling network underlying diabetic complications, a network pharmacology approach to explore active compounds and their pharmacological mechanisms may help to generate and apply the necessary knowledge to move the field of ROS towards mechanism-based, precision medicine [46].

## 4. ROS and Antioxidants in Diabetic Complications: Pitfalls and Stumbles on the Road to Therapeutics

Despite the promising preclinical evidence, it remains puzzling why beneficial effects of antioxidants have not yet been translated into the clinic. Several ROS sources have been identified and intensively studied in the last two decades. Various antioxidant-based intervention strategies that target specific ROS sources appeared promising in experimental DM models, but this research has not led to any antioxidant drug entering the routine clinical practice until now. What is more, the failure of clinical trials to prove the efficacy of antioxidants against diabetic complications has contributed to challenge the view of oxidative stress as the main mediator of diabetic complications. In fact, except for a study suggesting a therapeutic potential of lipoic acid by improving endothelial dysfunction in diabetic patients [76], classical antioxidants, including vitamin E, vitamin C, and N-acetylcysteine have always failed to show a significant protective effect, possibly due to the short duration of the studies [77,78,79]. However, even long-term supplementation with vitamin E was unable to reduce major cardiovascular events in diabetic patients in a large phase III trial such as the Heart Outcomes Prevention Evaluation (HOPE) study [80]. The reasons for this failure are many and varied, including lack of important pharmacokinetic evaluations. In fact, these and other studies did not monitor plasma levels of the supplemented antioxidant, thus precluding the possibility to determine drug safety and effective drug range. In general, however, the reasons can essentially be traced back to the incomplete understanding of the pathophysiology of ROS in health and disease.

For long viewed as metabolic by-products or waste and simply mediators of a global redox disequilibrium in oxidative stress-related diseases, ROS also play critical roles in normal cellular function, including differentiation, proliferation, and repair [81]. For example, both nitric oxide generated by NOS enzymes and H_2_O_2_ produced by NOX4 act as messenger molecules contributing to vasodilation, proliferation, and other important cellular functions [82,83,84]. The recent advances in redox biology indicate that the impact of ROS depends not only on their quantities but also on the type of ROS produced and the site of their generation: tissue, cellular, and even subcellular location [85]. Together with the notion that each ROS specifically regulates several physiological functions, this may explain why systemically applied exogenous antioxidants may not necessarily have beneficial effects or even cause harm [86]. Furthermore, issues concerning physicochemical (i.e., solubility in aqueous solutions, lipophilicity, etc.), pharmacokinetic (i.e., bioavailability/frequency of administration) and pharmacodynamic (i.e., therapeutic ratio and onset and duration of action) properties have also been raised to explain the failure of antioxidants in human studies [87]. Finally, less restrictive inclusion criteria compared with preclinical studies and possible different stages of disease progression or clinical manifestation, may contribute to explain the poor success of antioxidants in clinical trials.

To overcome the lack of specificity of classical antioxidants about ROS sources and types and the possible negative effect of inhibiting important physiological functions of ROS, new therapeutic approaches are under investigation. In addition to pharmacological inhibition of the specific ROS sources discussed in Section 3, novel selective approaches aimed at upregulating endogenous antioxidant enzymes have been proposed (Figure 4).

This indirect antioxidant strategy follows the suggestion that relevant therapeutic benefit may be achieved by favoring the activity of antioxidant enzymes rather than providing the antioxidant molecules. This approach is aimed at enhancing the physiological antioxidant response by providing molecules that act as agonists of transcription factors regulating the expression of antioxidant and cytoprotective genes, rather than exerting a direct antioxidant effect. These include synthetic chemicals, such as bardoxolone methyl and dimethyl fumarate (the ester derivative of fumaric acid), and natural compounds, such as sulforaphane obtained from cruciferous vegetables, which stimulate the master regulator of cell homeostasis nuclear factor (erythroid-derived 2)-like 2 (NRF2) [85,88], and inducers of the glyoxalase system, such as trans-resveratrol [89]. These pharmacological agents, generally referred to as indirect antioxidants, can assist the detoxifying process in vivo by inducing the expression of both endogenous antioxidant enzymes, including SODs, catalase, and glutathione peroxidase, and detoxifying enzymes, such as the components of the glyoxalase system that hydrolyze RCS and other ROS-derived toxic by-products [85]. In particular, the NRF2 activator bardoxolone methyl was confirmed to improve kidney function in a phase III clinical trial (ClinicalTrial.gov registration No. NCT01351675) conducted on T2DM patients with stage 4 DN, but the study was interrupted because of an increased number of cardiovascular events in the treated arm due to fluid retention [90].

Finally, given the importance of subcellular localization, using targeted delivery approaches to direct antioxidants to the site of ROS generation may be a viable option to overcome the current issues with antioxidant drugs in DM (see Section 5).

In summary, in view of the puzzling results obtained with classical antioxidants compounds, alternative therapeutic strategies aimed at interfering directly with one or more specific ROS-producing enzymes, possibly at the site of their production, and the indirect antioxidant strategy aimed at upregulating the physiological antioxidant defense system, seem to be more promising therapeutic options for diabetic complications.

## 5. Evidence against Mitochondrial ROS Overproduction as Initial Mediator of Glucotoxicity

An alternative explanation for the failure of clinical trials to demonstrate a protective effect of classical antioxidants on diabetic complications may be pathophysiological in nature. In this regard, an important contentious issue is whether oxidative stress occurs at an early stage in DM, preceding the onset of complications, or whether it is the result of cell and tissue damage, merely reflecting the presence of complications. Moreover, whether the activation of pathological pathways is upstream or downstream to ROS overproduction is an important point that need to be clarified. In fact, in the first case, the intervention with antioxidants could be potentially useful for managing the progression of complications, but it would have no effect on the early and intermediate steps of the pathogenic process.

The hypothesis that mitochondrial superoxide generation is the master upstream mediator of diabetic complications by increasing polyol and hexosamine pathway flux, protein kinase C activation, and AGE formation is frequently regarded as a paradigm even today. However, in the last fifteen years, several results have challenged this hypothesis. Studies conducted in genetically modified mice lacking the p66 kDa isoform of the Shc adaptor molecule, a master regulator of cellular ROS status, demonstrated that hyperglycemia induces both mitochondrial and cytosolic ROS production [91] and that ROS generation is, at least in part, a downstream event of AGE formation triggered by receptor-mediated events, including NOX4 transactivation [92]. In fact, in addition to provide protection against hyperglycemia-induced damage, genetic ablation of p66Shc also protected from DM-like glomerular disease induced by administration of preformed AGEs to euglycemic mice [92], which excluded a role for glucose-induced production of mitochondrial ROS.

More direct and compelling evidence against the hypothesis of a primary role of mitochondrial ROS has been gathered recently from preclinical studies showing that T1DM and T2DM vascular complications can occur in the absence of excess superoxide, or even in the presence of reduced kidney and cardiac superoxide levels, which was associated with mitochondrial dysfunction. Consistently, enhancement of mitochondrial biogenesis and oxidative phosphorylation led to increased superoxide production with beneficial effects on kidney function and fibrosis [93]. In addition, the levels of mitochondrial-derived metabolites and mitochondrial biogenesis were found significantly reduced in urine and kidneys of diabetic patients with DN [94,95]. Finally, treatment with the mitochondria-targeted drug elamipretide [96] or activation of 5’ AMP-activated protein kinase by 5-aminoimidazole-4-carboxamide-1-β-D-ribofuranoside [97] were found to protect against experimental DN by preserving physiological superoxide levels and mitochondrial function. These findings are at variance with data from other experimental studies demonstrating that treatments with new generation antioxidants targeting mitochondrial ROS, including coenzyme Q10 (CoQ10) [98,99] and mitoubiquinone mesylate (MitoQ) [100], have beneficial effects on DN. Reconciling these discordant results is difficult other than hypothesizing that opposite changes in mitochondrial ROS levels may play a specific role, or reflect different changes, at different stages in the sequence of the biochemical, molecular, and bioenergetic events that lead to the development of diabetic complications.

In summary, to identify efficient antioxidant-based therapeutic strategies, there is first a need to reassess the physiological and pathological relevance of ROS, and then to determine whether an antioxidant approach is feasible, in which situations, and if an optimal timing for treatment exists in the natural history of the disease [85]. In general, however, biochemical and molecular studies are warranted to investigate novel putative pathogenic mechanisms leading to the development of vascular complications. This effort might open up new therapeutic avenues in end-organ damage prevention and the maintenance of a healthy vasculature.

## 6. Alternative Mechanisms Linking Hyperglycemia to Cell Injury and Diabetic Complications

As discussed above, although the role of mitochondrial superoxide production remains controversial, there is convincing evidence showing that mitochondrial dysfunction and concomitant alterations in cellular energy production are critical mediators of diabetic complications. From this point of view, ROS dysregulation may be an epiphenomenon of permanent cellular bioenergetic and biochemical dysfunction, without direct therapeutic relevance to the pathogenesis of diabetic complications. To explore this hypothesis, alternative mediators of gluco- and lipotoxicity as well as mechanisms underlying the development and progression of diabetic complications have been under investigation, particularly in the past 10 years. In this section, we discuss two alternative mechanisms—namely carbonyl stress and the Warburg effect—for the pathogenesis of diabetic complications. As in Brownlee’s hypothesis, these mechanisms involve quantitative and qualitative changes in cellular glucose metabolism and abnormal mitochondrial function, but they focus on different key culprits, other than ROS, in the activation of the pathogenic pathways known to lead to vascular cell injury and inflammation.

### 6.1. Carbonyl Stress

Carbonyl stress is a condition characterized by a generalized increase in the steady-state levels of reactive carbonyl compounds, namely RCS. Replacing the nature of the reactive species (i.e., ROS instead of RCS), the same definition also applies to oxidative stress. The carbonyl stress hypothesis is inextricably intertwined with the oxidative stress hypothesis, as some RCS require oxidative reactions for their formation (Figure 5).

Actually, however, compared with oxidative stress, carbonyl stress is a more comprehensive term for a whole number of reasons. Firstly, RCS steady-state concentration is determined by the relative rates of their production and detoxification. Accordingly, chronic overload, exhaustion, or a decrease in the efficiency of the enzymatic pathways for RCS detoxification, including aldehyde dehydrogenases and the glyoxalase pathway, may concur to their accumulation [101]. Importantly, although some RCS originate from oxidative modifications of sugars and lipids (i.e., glycoxidation and lipoxidation reactions), other RCS are normal by-products of cellular glucose metabolism. Therefore, glucose excess may contribute to carbonyl stress in DM by causing an overproduction of glycolysis-derived RCS. Finally, elevated levels of oxidizable substrates such as glucose and lipids (i.e., substrate stress) is sufficient to explain the increase not only of RCS derived from glycolysis, but also of RCS originating from oxidative modifications, without the need of invoking an increase of ROS formation [71]. Considering all this, oxidative stress, intended as an increase in ROS levels, is dispensable for the initial build-up of RCS in DM. However, excess of oxidizable targets (i.e., substrate stress) may alter redox signaling and balance (i.e., cause redox dysfunction) by competing for ROS.

Glycolysis-derived RCS include the α-dicarbonyl compounds 3-deoxyglucosone (3DG) and MGO. These RCS are formed by the spontaneous degradation of triosephosphate intermediates of glycolysis (Figure 2). They are a physiological side-product of glucose metabolism and do not require oxidative reactions for their formation. In insulin independent cells like endothelial, neuronal and renal cells, glycolysis-derived RCS are produced at an increased rate because of hyperglycemia-induced intracellular metabolism of excess glucose [102,103,104,105,106]. Similar to other RCS, by reacting with free amino groups and thiol groups, glycolysis-derived dicarbonyls induce physico-chemical changes of proteins and nucleic acids that affect many functions of these biomolecules, including half-life, enzymatic activity, ligand binding, and immunogenicity (“direct damage” in Figure 4) [107,108]. MGO- and 3DG-mediated protein glycation leads to the formation of numerous AGE structures, such as MG-H1, 3DG-H1, and argpyrimidine, and protein crosslinking structures, such as the MGO-derived dilysine imidazolium cross-links [109,110,111]. In addition to exerting direct effects, these RCS can induce indirect biological effects through binding of AGEs to receptors of the innate immune system and induction of a chronic inflammatory response [112]. In particular, AGE binding to RAGE activates the transcription factor NF-κB, which regulates hundreds of genes involved in cellular stress responses, including redox sensitive signaling pathways leading to ROS formation, inflammation, and fibrosis (“indirect damage” in Figure 4) [113,114]. Accordingly, besides representing reliable biomarkers of carbonyl stress and tissue damage [71], AGEs are thought to contribute to the development of diabetic vascular complications [115].

Convincing evidence in favor of a central role of MGO in endothelial dysfunction and the pathogenesis of vascular complications has been obtained in preclinical models. Endothelial overexpression of glyoxalase 1 (Glo1), the rate-limiting enzyme of the glyoxalase system, was demonstrated to completely prevent hyperglycemia-induced formation of the major AGEs CML and N(ε)-(carboxyethyl)lysine (CEL) [102] and to reduce AGE accumulation in diabetic rats [116], indicating MGO as an important AGE precursor in endothelial cells. Reduced AGE formation in Glo-1 overexpressing rats was associated with improved DM-induced impairment of vasodilatation [117,118] and reduced expression of markers of endothelial activation [118]. In agreement with findings on vascular endothelial dysfunction, overexpression of Glo1 has been shown to improve DN by reducing albuminuria [118,119,120] and podocytes loss [118], two major pathological hallmarks of DN [121]. Finally, the causal role of MGO in endothelial dysfunction and DN is supported by the fact that exogenous administration of MGO to rodents has shown to induce DM-like vascular and renal changes, including impaired vasodilation [122,123] and glomerulosclerosis [124].

RCS derived from oxidative modifications of carbohydrates and lipids (i.e., glyco- and lipoxidation products) include the unsaturated trialdehyde acrolein and the dycarbonyl glyoxal (GO), originating from both glucose and lipids, and malondialdehyde and 4-hydroxynonenal, deriving only from lipids [125,126] (Figure 5). These RCS are four major and biologically active aldehydes, and similar to glycolysis-derived RCS their levels are increased in DM [127,128,129]. However, considering that substrate stress-derived RCS, particularly glycolysis-derived RCS, can account for the initial increase of carbonyl stress burden in DM [130], RCS derived from increased ROS production likely play a role in more advanced stages, when accumulation of AGEs and AGE-mediated activation of pro-inflammatory/oxidative pathways have already occurred. Accordingly, intervention should be directed at reducing carbonyl stress, long before the occurrence of overt oxidative stress and damage. Antioxidant therapy may not only be late but also miss a large fraction of the target that bring about tissue damage (i.e., nonoxidative-derived RCS, AGE, and RAGE activation). Therefore, an efficient therapeutic strategy should be directed at reducing AGE formation by trapping RCS originated from excess glucose and lipids, regardless of whether they are originated from oxidative stress (i.e., increased ROS production) or not.

Strategies to lower RCS, including MGO, have been developed over the past years, the most important of which are RCS scavengers and glyoxalase inducers. RCS trapping agents, including hydrazine derivatives, such as aminoguanidine, vitamin B derivatives, such as pyridoxamine, and amino acid derivatives, such as *N*-acetyl cysteine and L-carnosine, have been tested with encouraging results in preclinical studies, as comprehensively reviewed before [108,131]. Among the novel compounds, the carnosinase-resistant l-carnosine derivatives carnosinol and the enantiomer D-carnosine were shown to be effective in attenuating vascular complications of both DM [132,133] and dyslipidemia [134]. A suitable alternative may be the enhancement of RCS detoxification with active drugs targeting the glyoxalase system or the master regulator of cell homeostasis and inducer of phase II detoxification enzymes NRF2 [87,88]. Targeting carbonyl stress, particularly the early MGO burden, may provide new therapeutic opportunities to mitigate diseases in which this reactive dycarbonyl plays a critical role, such as diabetic complications.

In summary, as in the unifying hypothesis by Brownlee and colleagues, the core of the problem seems to be the handling and disposal of excess fuel by the cell. Consistent with a critical role of excess glucose flux through the glycolytic pathway in hyperglycemia-induced carbonyl stress, increased MGO formation and accumulation was also observed in activated inflammatory cells and hypoxia, two conditions characterized by boosted aerobic glycolysis and reduced mitochondrial respiration [135,136,137,138]. Interestingly, the shift from oxidative phosphorylation to the glycolytic pathway (i.e., aerobic glycolysis) is a hallmark of the Warburg effect, another mechanism deemed to play a role in the very early stages of diabetic complications.

### 6.2. Warburg Effect

Over the last two decades, a growing body of evidence has shown that diabetic complications are associated with altered mitochondrial function, as well reviewed in recent articles [139,140,141]. Several groups have demonstrated that reduced mitochondrial function plays a critical role in DN, both in preclinical models and in DM patients [93,96,142]. Unfortunately, the reason mitochondria are dysfunctional in DM is still far from clear. The acquisition of this information is decisive for defining a pharmacological strategy aimed at repairing mitochondrial damage, correcting mitochondrial dysfunction, and verifying its effectiveness in preventing chronic complications. It is now clear that DM alters the delivery of nutrients to the tissues that results in switching to alternative substrates for cell energy production [139]. For example, substantial evidence indicates that enhanced mitochondrial fatty acid oxidation is involved in diabetic complications [143]. What is more, recent omics studies have shown that mitochondrial dysfunction, as defined by suppressed mitochondrial activity and biogenesis, plays a crucial role in the development of human DN and is associated with a metabolic rewiring that resemble the Warburg effect [93,143,144].

Mainly investigated in the field of cancer biology, the Warburg effect is defined as an increase in the rate of glucose uptake and preferential fermentation of glucose to lactate, even in the presence of oxygen. The Warburg effect is a feature of glycolytic cancer cells, which are so-called because they tend to favor metabolism via aerobic glycolysis rather than oxidative phosphorylation [138,145,146,147]. However, a switch to aerobic glycolysis resembling the Warburg effect has been also involved in non-cancer diseases, including in the process of inflammation. In fact, a reduction of glucose oxidation and a shift towards a glycolytic metabolism are required for macrophage [136,137,148] and T cell [148] activation.

The Warburg-like metabolic reprogramming described by Sharma et al. in human DN [93] has been further investigated and confirmed in a preclinical model of T2DM by using a system biology approach combining transcriptomic, metabolomic, and metabolic flux analysis [143]. This study showed that mitochondrial metabolic alterations are paralleled by an enhanced metabolic flux into glycolysis, as attested by an increase of several glycolytic enzymes, including hexokinase, phosphofructokinase, and pyruvate kinase [143]. Several metabolic intermediates, including sphingomyelin, TCA metabolites such as fumarate, and glycolytic enzymes such as the M2 isoform of pyruvate kinase (PKM2) have been proposed as potential regulating factors in shifting glucose from complete oxidation into the glycolytic pathway and its principal branches [149]. However, what is the pathophysiological meaning of these changes in mitochondrial function and bioenergetics and what are the early instigator(s) of the Warburg effect in DM have not yet been clarified. In particular, it has not been established if a cause–effect relationship exists between mitochondrial dysfunction and the Warburg-like effect in DM, how these biological processes cooperate in the onset of DM end-organ damage, or if and how they influence each other. Surprisingly, despite enhanced glucose uptake and mitochondrial and glycolysis dysfunction have long been described in diabetic tissues [21], the Warburg effect as a feature of diabetic complications was not proposed for a long time. The reasons for this delay are unclear but may concern inadequate cross-disciplinary communication and lack of a holistic perspective.

To shed some light on the possible role of the Warburg effect in diabetic complications, it is necessary to go through this complex and not completely understood bioenergetic phenomenon in a bit more detail. As mentioned above, this metabolic reprogramming has been extensively investigated in cancer cells since the 1920s, when professor Otto Warburg showed that, even under aerobic conditions, most tumor tissues continued to produce elevated levels of lactate [149,150,151]. Consistent with a physiological relevant role of the shift from mitochondrial respiration to glycolysis, a positive relationship between the rate of glucose metabolism and that of cell growth has been demonstrated in glycolytic cancer cells [150]. Accordingly, withdrawing glucose or inhibiting glycolysis is deleterious to tumor cell growth and tumorigenesis in experimental models [152,153]. However, it is still not fully understood what are the molecular mechanisms by which the Warburg effect confers a selective advantage to cancer cells. One of the factors at play is that the increased flux of glucose through the glycolytic pathway allows cancer cells to maintain large pools of glycolytic intermediates to sustain anabolic metabolism by feeding several non-mitochondrial biosynthetic pathways that branch from glycolysis [154,155] (Figure 6).

Consistent with this function, glycolytic cancer cells also express high levels of the M2 isoform of the glycolytic terminal enzyme PKM, which has emerged as a key factor in the regulation of aerobic glycolysis [156,157]. At variance with the constitutively active PKM1 isoform, PKM2 is mainly present in a dimeric, less active form. High ratios of PKM2/PKM1 boost glycolysis and decrease the glucose flux into oxidative phosphorylation [158], thus favoring accumulation of upstream glycolytic metabolites [159]. As a proof of the critical role of PKM2, switching PK expression to the M1 isoform, or the pharmacological activation of the PKM2 isoform (i.e., induction of tetrameric PKM2), leads to reversal of the Warburg effect in glycolytic cancer cells [160]. Having full pools of glycolytic intermediates favors cancer cells to engage metabolic branch pathways of glycolysis, including hexosamine pathway, which is required for protein glycosylation, and pentose phosphate pathway, which produces ribose for nucleotides and NADPH synthesis. Moreover, increased levels of the triose phosphates favor glycerol production for the synthesis of complex lipids and for serine/glycine biosynthesis and one-carbon metabolism. Altogether, these pathways supply the necessary substrates for the synthesis of proteins, nucleic acids, and lipids and the maintenance of the redox homeostasis (e.g., glutathione reactions) [155]. It is important to note that, in cancer cells, these pathways are activated in response to oncogenic signaling.

Interestingly, the levels of mitochondrial and glycolytic enzymes, particularly PKM2, were recently associated with the susceptibility to DN in T1DM and T2DM patients [144,161]. In agreement with a role of PKM2 and the Warburg effect in diabetic complications, pharmacological activation of the enzymatic activity of PKM2 was recently associated with improved mitochondrial function and biogenesis, suppressed lactate accumulation, blunted elevation in glycolytic intermediates and glycotoxins, and reduced renal inflammation and fibrosis in experimental models of DN [144,162,163]. A necessary observation is that, although activated through different mechanisms, the process of diversion of glycolytic intermediates into the glycolytic branch pathways that benefits cancer cells is the same that has been claimed to be responsible for the deleterious effects of hyperglycemia in normal cells [24,149] (Figure 2). In simple words, what is good for cancer cells is bad for normal cells. However, this should not be surprising, as cancer cells contain mutations that lead to unchecked cell growth and proliferation, whereas normal cells move through the cell cycle in a regulated manner. Accordingly, in cancer cells, pathways branching from glycolysis efficiently use glycolytic intermediates to ensure the availability of essential building blocks to sustain cell proliferation and tumor growth. In normal cells, enhanced glucose entry and glycolysis and the resulting unnecessary diversion of glycolytic intermediates into the same pathways eventually induce the accumulation of downstream metabolites leading to the activation of biochemical and signaling networks that are known to cause damage, inflammation, and fibrosis in DM [2,164]. Therefore, hyperglycemic damage in normal cells may be related to their limited ability to use the surplus of glucose metabolic intermediates for growth purposes. In this regard, it is interesting to note that the initial and transient response of mesangial cells in DN is just proliferation, followed by cell cycle arrest, hypertrophy, and activation of the proinflammatory and profibrotic pathways that lead to over secretion of extracellular matrix proteins and consequent glomerular remodeling. [165]. What is more, a hyperplastic response to hyperglycemia is an early and transient event that has also been observed in other cell types (i.e., smooth muscle cells and endothelial cells) playing a role in the damage of other tissues, including the arterial wall [166] and retina [167].

Altogether these findings pave the way for further research into the link between carbonyl stress, mitochondrial dysfunction, and the Warburg effect in diabetic complications. They also suggest potential avenues for prevention and treatment of end-organ damage induced by DM. From a mechanistic perspective, the glycolysis by-product MGO has now been recognized as a potential driver for both the development of diabetic complications [130,168] and the progression of several highly glycolytic cancers [73]. In addition, a metabolic rewiring similar to that of glycolytic tumor cells has recently been described as a feature of diabetic complications [144,149]. Therefore, to further improve our understanding of the etiology of diabetic complications and identify key mediators of glucose toxicity, the interplay between carbonyl stress and aerobic glycolysis (i.e., Warburg effect) warrants focused investigation.

## 7. Conclusions

Oxidative stress, inflammation, and the resulting tissue damage are hallmarks of chronic diseases, and DM is not an exception. Most of the literature includes oxidative stress as an important element in the pathogenesis of diabetic complications. However, because of new knowledge on the pathophysiology of ROS in health and disease, the classical view of oxidative stress as a quantitative deviation from an overall equilibrium of ROS formation and cellular antioxidant defense has changed into a more complex phenomenon that implies considering the chemical nature of ROS, subcellular or tissue location, and chemical kinetic mechanisms [169]. However, the real point of discussion is not so much whether oxidative stress, however defined, is increased in DM, but whether oxidative stress plays an initiating role in glucotoxicity and the pathogenesis of diabetic complications. This issue is important in terms of identifying an early therapeutic target capable of preventing the development and blocking the progression of DM-related chronic complications. Therefore, the question is whether ROS are the sparks that ignite the fire or, rather, are the fire flames.

Randomized controlled trials in which antioxidants were given with standard DM treatment have shown inconsistent efficacy results [76,77,78,79], which can be compared to dropping a bucket of water on a wildfire. It could be helpful in limiting the damage, but also late in the course of conflagration. Several reasons may explain the failure of classical antioxidants in the clinical setting, including the poor pharmacokinetic and pharmacodynamics of the tested compounds and issues related to concentration, specificity, and time and stage of administration [85]. To overcome these shortcomings, new approaches with direct antioxidants aimed at interfering with specific ROS-producing enzymes, such as NOX inhibitors, or with indirect antioxidants that target redox enzymes, such as NRF2 agonists, are under intensive clinical investigation. Things are further complicated by recent experimental studies suggesting that ROS are dispensable for the initiation of tissue damage induced by DM and, hence, that excess ROS is not the primary instigator of diabetic vascular complications [92,95].

These findings compel us to search for alternative triggers of the cellular and biochemical abnormalities that are widely recognized as the major pathological factors involved in diabetic complications: i.e., mitochondrial dysfunction, the activation of pathways of glucotoxicity (particularly RCS and AGE formation), inflammation, and redox imbalance. Similar to Brownlee’s hypothesis, two promising lines of investigation concern the changes in cellular glucose metabolism induced by hyperglycemia in target tissues of diabetic complications. Both the increased formation of the glycolytic by-products and highly reactive α-oxoaldehydes (i.e., carbonyl stress) and the establishment of a Warburg effect are closely related with intracellular metabolism of excess glucose and mitochondrial dysfunction, but independent from oxidative stress. It is intuitive and there is evidence of an interaction among quantitative/qualitative changes in cellular glucose metabolism, RCS formation, mitochondrial dysfunction, and the Warburg effect, but it is unknown whether the Warburg effect and carbonyl stress are independent processes with an underlying common cause or are linked by a cause–effect relationship. These are open questions which should be investigated to search for an alternative unifying mechanism of diabetic complications.

To develop more effective preventive and therapeutic strategies, identifying the spark that makes the fire flare up and spread should be the goal of our research from tomorrow onwards.

## Figures and Tables

**Figure 1 antioxidants-10-00727-f001:**
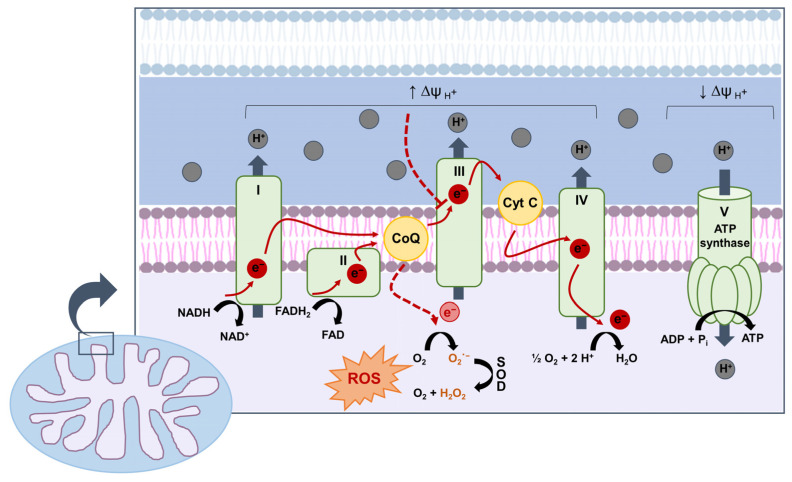
Production of reactive oxygen species (ROS) by mitochondrial electron transport chain (ETC). The electron donors NADH and FADH2 generated by the oxidation of glucose-derived pyruvate feed electrons (e^−^) into Complexes I and II of the ETC, respectively. Traversing the ETC of the inner mitochondrial membrane, e^−^ release some of their energy to pump H^+^ into the intermembrane space, generating a voltage gradient (∆ψ) that is used to move H^+^ through ATP synthase and phosphorylate ADP to ATP. When the ∆ψ reaches a critical threshold due to increased flux of electron donors, e^−^ transfer from coenzyme Q (CoQ) to Complex III is hindered and more superoxide (O_2_^−^) is generated (red dashed lines). In turn, O_2_^−^ dismutation catalyzed by superoxide dismutase (SOD) generates hydrogen peroxide (H_2_O_2_). Cyt C, cytochrome complex; ROS, reactive oxygen species.

**Figure 2 antioxidants-10-00727-f002:**
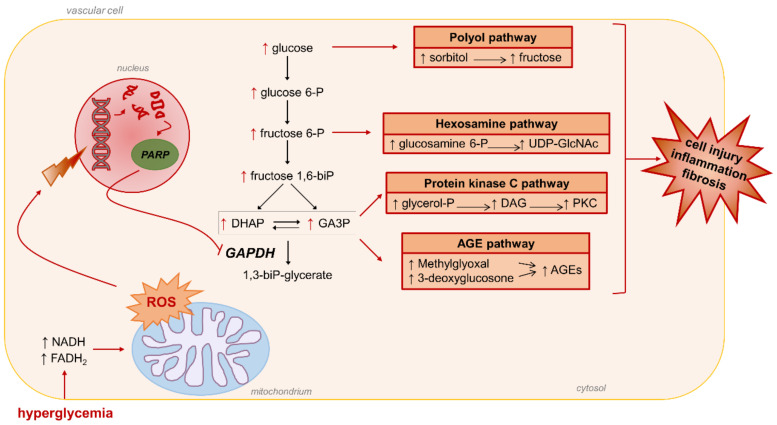
Brownlee’s unifying mechanism for hyperglycemia-induced diabetic complications. Mitochondrial ROS-induced DNA damage activates poly(ADP-ribose) polymerase (PARP) that modifies and reduces the activity of glyceraldehyde 3-phosphate dehydrogenase (GAPDH). The accumulation of glycolytic intermediates upstream GAPDH activates the four major pathways of hyperglycemic damage. AGEs, advanced glycation end-products; DAG, diacylglycerol; DHAP, dihydroxyacetone phosphate; GA3P, glyceraldehyde 3-phosphate; PKC, protein kinase c; ROS, reactive oxygen species; UDP-GlcNAc, uridine diphosphate N-acetylglucosamine.

**Figure 3 antioxidants-10-00727-f003:**
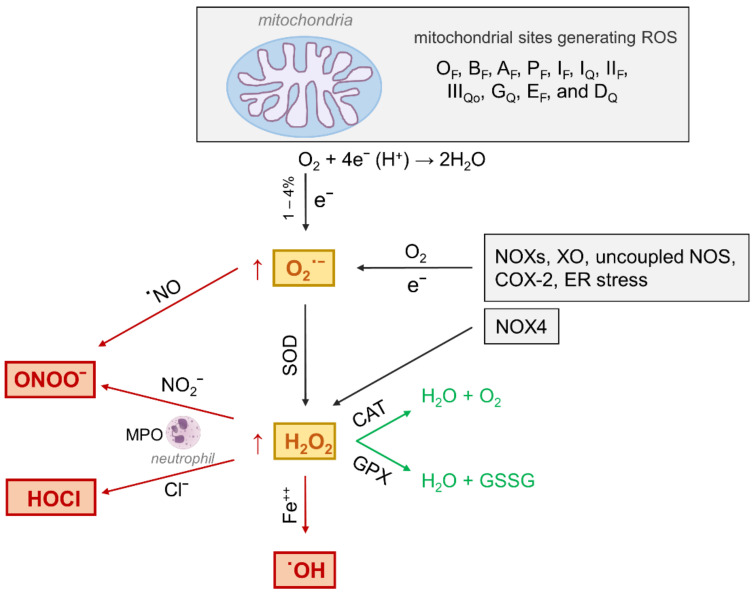
Oxidant species generating pathways and metabolism under physiological (green arrows) and oxidative stress (red arrows) conditions. The primary type of ROS produced in the body is superoxide (O_2_^−^), which is formed from single electron reduction of molecular oxygen (O_2_). Eleven mitochondrial sites generating ROS by leaking electrons to oxygen during substrate oxidation have been identified [40]. Beside mitochondria, NAD(P)H oxidases (NOXs), xanthine oxidase (XO), uncoupled nitric oxide synthase (NOS), cycloxygenase-2 (COX-2), and endoplasmic reticulum (ER) stress, contributing to O_2_^−^ production. In turn, superoxide dismutase (SOD) converts O_2_^−^ into hydrogen peroxide (H_2_O_2_). In addition, H_2_O_2_ can be directly produced by NOX4. Whatever its source, under physiological conditions, H_2_O_2_ is reduced by catalase (CAT) into oxygen and water and by glutathione peroxidase (GPX) into oxidized glutathione (GSSG) and water. However, when produced at increased levels, H_2_O_2_ generates highly reactive and cytotoxic oxidants such as hypochlorous acid (HOCl) by neutrophil myeloperoxidase (MPO) and the hydroxyl radical (OH**^•^**) in the presence of catalytically active iron (Fe^++^). In addition, O_2_^−^ combines with nitric oxide (**^•^**NO) to form peroxynitrite (ONOO^−^), which can also be produced by H_2_O_2_ in the presence of nitrite (NO_2_^−^) via MPO-mediated reaction.

**Figure 4 antioxidants-10-00727-f004:**
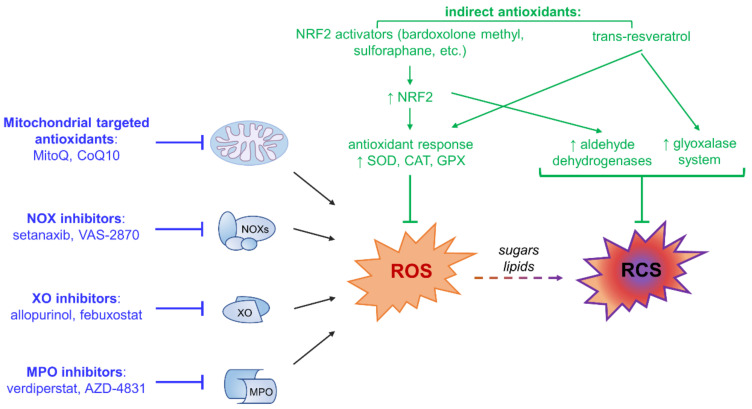
Pharmacological inhibition of specific reactive oxygen species (ROS) sources (blue) and indirect antioxidant strategies (green). Targeting specific sources of ROS may decrease oxidative stress by directly inhibiting ROS-producing enzymes, including NADPH oxidases (NOXs), mitochondria, myeloperoxidase (MPO), and xanthine oxidase (XO). Indirect antioxidants such as inducers of nuclear factor (erythroid-derived 2)-like 2 (NRF2) (i.e., bardoxolone methyl, sulforaphane, etc.) and glyoxalase system (i.e., trans-resveratrol) may help to maintain redox homeostasis by activating endogenous antioxidants and cellular detoxifying defenses against ROS and their by-products, including reactive carbonyl species (RCS). CAT, catalase; CoQ10, coenzyme Q10; GPX, glutathione peroxidase; MitoQ, mitoquinone; SOD, superoxide dismutase.

**Figure 5 antioxidants-10-00727-f005:**
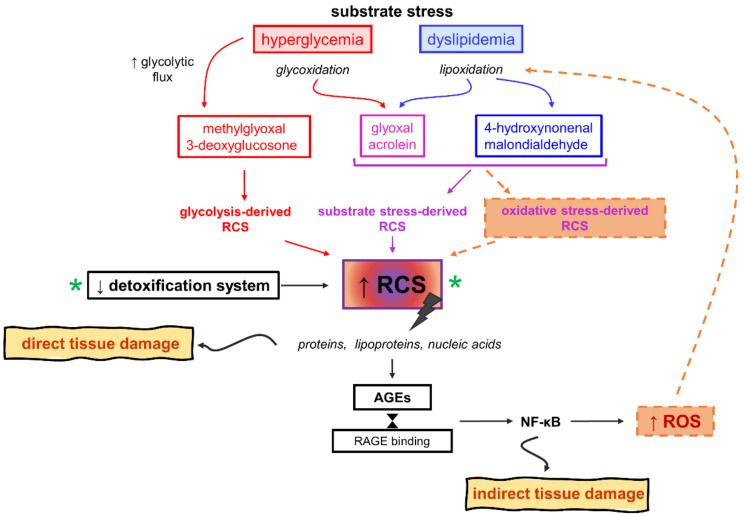
Carbonyl stress. Causes and mechanisms of reactive carbonyl species (RCS) and advanced glycation end-products (AGEs) formation, their relationships with oxidative stress, pathogenic role in tissue damage leading to diabetic complications, and potential drug targets (green asterisk). NF-κB, nuclear factor-κB; ROS, reactive oxygen species.

**Figure 6 antioxidants-10-00727-f006:**
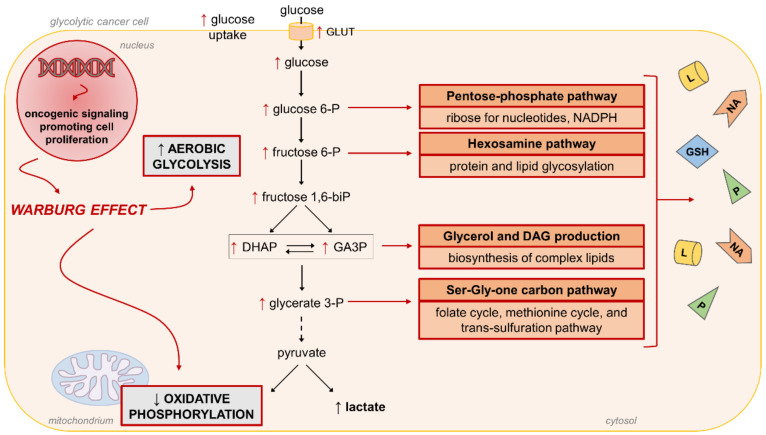
Warburg effect (aerobic glycolysis) and cell proliferation. Enhanced glucose flux through the glycolytic pathway supports cell growth by feeding several non-mitochondrial anabolic pathways that ensure the availability of essential building blocks to sustain cell proliferation and tumor growth. DAG, diacylglycerol; DHAP, dihydroxyacetone phosphate; GA3P, glyceraldehyde 3-phosphate; GLUT, glucose transporter.

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
