# Peer review of "Diabetic Complications and Oxidative Stress: A 20-Year Voyage Back in Time and Back to the Future"

_antioxidants, 2021, doi:10.3390/antiox10050727_

Round 1

Reviewer 1 Report

In the manuscript entitled “Diabetic complications and oxidative stress: a 20-year voyage 2 back in time and back to the future,” Iacobini et al. discuss the relationship between redox dysfunction and DM, the evidence for a causal relationship between ROS and DM complications, and the new theories implicating carbonyl stress and the Warburg effect as the culprits for DM complications. I want to congratulate the authors for a wonderful review. The review is clear, well organized and an excellent contribution to the field. I only have minor issues that I hope will help improve an already excellent work.

Minor issues

Why does section 2 and figure 3 show the other accepted sites of mitochondrial superoxide production?

Line 210 calls nitric oxide a ROS. However, later the authors say that NOXs are the only enzymes that evolved to produce ROS (not as a byproduct). If you call nitric oxide a ROS, then NOS enzymes also specifically produce ROS. So either call nitric oxide a reactive nitrogen species, or a reactive species (not ROS), or state that both NOXs and NOSs both produce ROS.

Figure 3 is confusing, as it appears as if MPO produces nitrite and chloride, not that MPO uses nitrite to produce ONOO-, and chloride to produce HClO.

The caption of figure 3 should say reactive species generating pathways instead of ROS generating pathways, as ONOO- and HClO (and nitric oxide) are included.

Please add the dot (free radical indicator) to nitric oxide in figure 3

In line 242 the authors state that NOX4 was formely termed Renox. Please state this the first time you introduce NOX4, and not here.

Section 4 discusses how the opposing effects of different ROS sources and the different impact of ROS in different organs might account for the disappointing clinical trial results. Given that localization is so important, I suggest the authors comment on targeted delivery approaches as a possibility to overcome the current issues with antioxidant drugs in DM

Line 316 mentions the HOPE trial. It should be discussed that the dose of vitE was not sufficient to exert antioxidant effects. Moreover, throughout the review this possible explanation for failed antioxidant trials is not discussed. The clinical trials measure clinical outcomes and the majority do not assess that the actual antioxidant effect was achieved. So, that an antioxidant doesn’t work could mean that ROS do not have a causal relationship with the pathology, or that the intervention did not actually changed ROS levels

Figure 4 has labels classes of drugs and examples (eg XO inhibitors: allopurinol, febuxostat). However, for Nrf2 activators it just says bardoxolone, not NRF2 activators: bardoxolone methyl. I specifically mention this because other Nrf2 activators have shown promising effects in preclinical models. Make the labels consistent within the figure.

In lines 461-465 it is discussed how carbonyl stress does not need oxidative stress. Even though I agree that it does not need an increase in ROS levels, if the oxidizable targets increase and compete for ROS, they will induce redox dysfunction by possibly altering redox signaling. Therefore, I argue that RCS derived from ROS-mediated oxidation, could potentially cause redox dysfunction.

Line 464 states “increase of ROS formation (i.e., oxidative stress).” This statement equated ROS increase with oxidative stress. It’s surprising that the authors wrote this here since the show an excellent understanding of the complexities of redox dysfunction. So, the authors clearly know that redox dysfunction is not synonym with ROS increase. Please modify the sentence accordingly.

 In line 690 the authors say that because ref 91 and 92 show that ROS are dispensable for initiating tissue damage, ROS must not be the initiator. I appreciate the evidence against ROS causal relationship with the initiation of damage. However, this new evidence is also in preclinical models and as the authors point out there is plenty of preclinical model evidence showing that ROS have a causal relationship with the initiation of damage leading to end-organ complications. Hence, I suggest changing the sentence to: “Things are further complicated by recent experimental studies suggesting that ROS are dispensable for the initiation of tissue damage induced by DM, and hence that excess ROS is not the primary instigator of diabetic vascular complications.”

During the discussion of carbonyl stress and the Warburg effect, the authors do not bring back the fact that even with strict glycemic control DM patients have worse vascular outcomes.

Reviewer 2 Report

Iacobini, Vitale and colleagues have developed a very comprehensive review on a highly relevant topic in diabetes research. The authors have pin-pointed many important points and discussed proposed mechanisms and potential pharmacological options to prevent and/or treat diabetic complications, namely related to vascular disease, related to ROS involvement.

The manuscript is very-well written and it was a great pleasure to read it. For that reason, only a few minor points are noted below.

1) In line 80, the specific Brownlee reference should be included (I guess it is [21]), as this author is also referenced in [6], for example.

2) In lines 103-5, it would be easier to the reader if 1) xxx, 2) xxx, etc would be added to clearly differentiate the 4 four major pathways.

3) There is an extra "." in line 122.

4) There is an extra "s" in "others" in line 459.

5) It is not clear why the authors are acknowledging a company for the gift of chemical compounds in a bibliographical review article.
